# Late-fall satellite-based soil moisture observations show clear connections to subsequent spring streamflow

Randal D. Koster [1] ✉, Qing Liu[1,2], Wade T. Crow [3] & Rolf H. Reichle [1]

Because runoff production is more efficient over wetter soils, and because soil moisture has an intrinsic memory, soil moisture information can potentially contribute to the accuracy of streamflow predictions at seasonal leads. In this work, we use surface (0–5 cm) soil moisture retrievals obtained with the National Aeronautics and Space Administration's Soil Moisture Active Passive satellite instrument in conjunction with streamflow measurements taken within 236 intermediate-scale (2000–10,000 km$^2$) unregulated river basins in the conterminous United States to show that late-fall satellite-based surface soil moisture estimates are indeed strongly connected to subsequent spring-time streamflow. We thus show that the satellite-based soil moisture retrievals, all by themselves, have the potential to produce skillful seasonal streamflow predictions several months in advance. In poorly instrumented regions, they could perform better than reanalysis soil moisture products in this regard.

Forecasting variations in streamflow at monthly to seasonal leads has obvious implications for water resources management. Because our ability to forecast precipitation at seasonal leads is marginal at best[1], accuracy in streamflow forecasts must generally rely on other facets of the climate system. Importantly, wintertime snowpack measurements, particularly in mountainous areas, provide significant information about streamflow during the subsequent melt season. As a result, snowpack estimates underlie many current streamflow forecast efforts[2,3].

A second facet of the climate system relevant to streamflow forecasting is soil moisture. Wet soils provide little pore space for infiltration and are thus particularly conducive to surface runoff production during storms; conversely, rainfall or snowmelt water incident on a drier soil is more likely to infiltrate and subsequently be lost to the atmosphere through later evapotranspiration (ET). Put simply, the soil's water content determines the degree to which the land surface is preconditioned to produce runoff from incident water. For seasonal streamflow forecasting, soil moisture has the additional advantage of having useful memory—a soil moisture anomaly at the start of a forecast can persist and thereby affect hydrological processes well into the

forecast period. This is particularly true during winter, when soil moisture anomalies persist longer due to lower ET rates[4]. Through the combined effects of this memory and the preconditioning mechanism, knowledge of soil moisture conditions on the start date of a seasonal streamflow forecast has the potential to contribute skill to that forecast[5,6], even in the absence of snowpack information or accurate seasonal rainfall forecasts[7].

In the present paper, we examine the ability of space-based soil moisture retrievals to provide the information needed to tap into this potential. The National Aeronautics and Space Administration (NASA) Soil Moisture Active-Passive (SMAP) mission[8] has provided global estimates of near-surface soil moisture (over a nominal depth of 5 cm) since the spring of 2015. Studies have already examined the connection between SMAP retrievals and runoff efficiency at synoptic time scales[9]; here, we investigate their relevance to runoff production at the seasonal time scale, at seasonal leads. We first utilize an exponential temporal filter (Methods) to infer moisture deeper in the soil from the surface retrievals. We then quantify the statistical connection between the inferred deeper soil moistures, as determined for late fall, and gauge-measured streamflow totals in the subsequent spring. Our

[1]Global Modeling and Assimilation Office, NASA Goddard Space Flight Center, Greenbelt, MD, USA. [2]Science Systems and Applications, Inc., Lanham, MD, USA. [3]U.S. Department of Agriculture, Agricultural Research Service, Hydrology and Remote Sensing Laboratory, Beltsville, MD, USA. ✉e-mail: randal.d.koster@nasa.gov

analysis demonstrates that the connection is indeed significant and that SMAP data therefore do have the potential to contribute skill to predictions of seasonal streamflow.

## Results

We present two sets of results in this section. In the first subsection, we describe the calibration of a parameter that allows us to infer estimates of deeper soil moisture from the surface moisture retrievals provided by SMAP. With this calibration in hand, we describe in the second subsection our main findings: the connection between the deeper soil moisture estimates so obtained and streamflow measured several months later.

### Calibration of $\tau$ parameter

The $\tau$ parameter describes the time scale over which we exponentially filter the surface soil moisture retrievals into estimates of profile moisture on November 30, $W_{Nov30}$ (see Methods). It is a key parameter for this research, a parameter that would presumably be relevant to

any study that attempts to infer deeper soil moisture from surface retrievals alone. Figure 1a shows the results of the $\tau$ calibration exercise: it shows, as a function of $\tau$, the spatial average (across the 236 basins) of the local Pearson's correlation coefficient $R$ between the SMAP-derived $W_{Nov30}$ and the MERRA-2 reanalysis's November 30 full profile soil moisture, an independent estimate of the soil moisture over a depth of 1.3–3.5 m, depending on location. For all $\tau$ values exceeding ~20 days, the spatial average exceeds 0.7, suggesting that any of these time scales could provide, with some skill, estimates of the interannual variation of profile soil moisture. The spatial average, however, has a maximum at $\tau = 38$ days. This is the time scale we will use in the next section to generate SMAP-based November 30 soil moisture estimates for correlation against subsequent springtime streamflow.

We should note, however, that while $\tau = 38$ days is an overall optimal time scale for the basins, it may be suboptimal in some areas. Figure 1b, c shows the $R$ distributions across the stream gauge sites for, respectively, the optimal $\tau$ value and $\tau$ equal to 7 days. As must be the case, the average $R$ is higher for $\tau = 38$ days ($R = 0.72$) than for $\tau = 7$ days

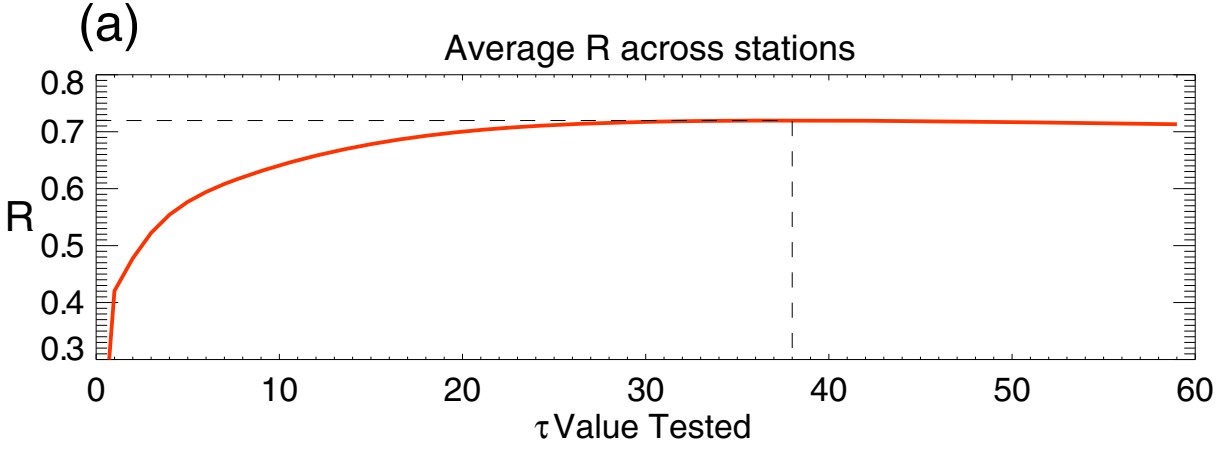

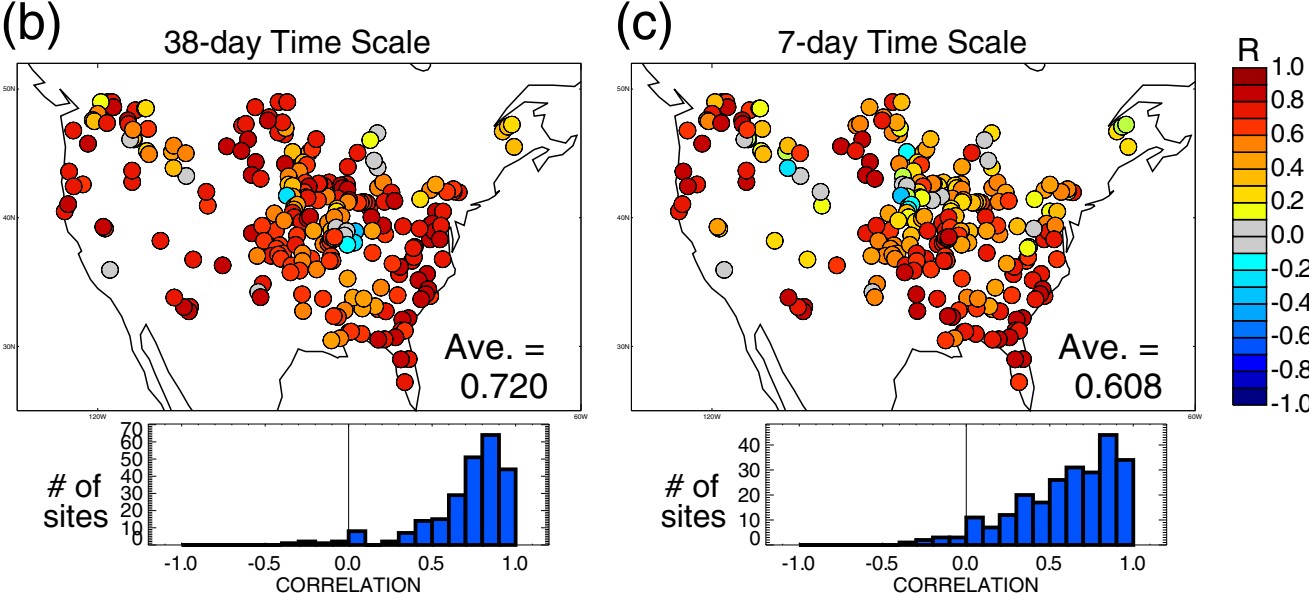

**Fig. 1 | Estimating November 30 profile soil moisture with Level 2 (L2) retrievals. a** Results of the time scale calibration exercise: spatially averaged correlation coefficient $R$ (that between Soil Moisture Active-Passive (SMAP)-estimated basin soil moisture for November 30 and corresponding profile soil moisture from the Modern-Era Retrospective analysis for Research and Applications, Version 2 [MERRA-2] reanalysis) as a function of the time scale, $\tau$, applied in Eq. (1) of

Methods. The basin-level $R$ values are averaged here across the 236 basins. **b** Local values of the correlation coefficient, $R$, underlying the spatial average in (**a**) for the optimal time scale of 38 days. For ease in visualization, values are plotted with circles of uniform size at the stream gauge locations. The attached histogram provides an overall summary of the results. **c** As in (**b**), but for a 1-week time scale.

($R = 0.61$). Indeed, $R$ values for $\tau = 38$ days are generally high across the continental United States (CONUS), supporting the time scale's general application. Nevertheless, $R$ values for some locations (e.g., in the lower Mississippi Basin) are higher for $\tau = 7$ days, a distinction that hints at the potential benefit of applying a spatially varying time scale. We leave a treatment of spatial variability in $\tau$ to future work, noting the difficulties that would be encountered in such a treatment given the small sample sizes—each local $R$ calculation in the calibration exercise is based on only 7 sample pairs. Again, here we will simply use the single value ($\tau = 38$ days) supported by the joint consideration of all 236 basins.

### Correlations between soil moisture and streamflow

Figure 2a shows the spatial distribution of the correlations between the estimated $W_{Nov30}$ values (using the calibrated $\tau$) and the measured streamflow rates accumulated over the subsequent spring (February–May) season. (See Supplementary Figs. S1–S3 for examples of soil moisture, streamflow, and precipitation time series in representative basins.) As already noted, a temporal correlation computed from seven data pairs is highly uncertain. Nevertheless, such a correlation can be considered significantly different from zero with 95% confidence if it exceeds 0.68 (Methods); this condition is met by 22% of the basins in Fig. 2a, well above the 5% of basins for which the criterion would, on average, be met by chance. Most importantly, the values shown in Fig. 2 are overwhelmingly positive, with negative values appearing for only 19 of the 236 basins examined. The probability of such a positive/negative breakdown occurring by chance is vanishingly small. The prevalence of positive values appears throughout CONUS, which is noteworthy given the hydrological diversity of the basins considered. Positive values dominate in both mountainous and nonmountainous regions and in both seasonally snow-dominated and snow-free regions.

Curiously, the few negative or very small correlations that do appear mostly lie in the northernmost parts of the study region. While some of the larger negative correlations reflect unexpected springtime precipitation extremes (Supplementary Fig. S3), we note that the more

northern areas are often snow-covered in late November. Arguably, snowpack present on November 30 represents water that could otherwise have infiltrated and modified the soil moisture before and up to that date; as a result, we can speculate that late fall snowpack has the effect of degrading the connection between $W_{Nov30}$ and subsequent streamflow. More analysis, of course, in conjunction with a longer data period would be needed to explain the correlations found at individual locations.

The raw average of the correlations shown in Fig. 2a is 0.43. A cluster analysis (Methods) provides some slightly different averages: using a cluster radius of 2 degrees, the average is 0.42, with a 95% chance that the true average lies between 0.29 and 0.49, and with a more conservative cluster radius of 3 degrees (i.e., an assumption of a larger spatial correlation length scale between the measurements), the mean becomes 0.38, with the 95% confidence interval ranging from 0.19 to 0.49. Overall, regardless of whether we focus on the raw correlations or the cluster analysis results, and despite the noted sample size limitations, the joint consideration of the results across the continent indicates that SMAP L2 soil moistures provide real information on streamflow totals at a seasonal lead.

Supplemental calculations can address the impact of retrieval reliability on the correlation analysis. Up to this point, our analysis has used SMAP retrievals with an "uncertain" quality flag along with those having a "recommended" flag (Methods, section "SMAP soil moisture retrievals"); this lax constraint allowed the basins we consider to better cover CONUS. If we now repeat the analysis above using a more limited set of SMAP retrievals—only those with the "recommended" quality flag—we obtain the revised results in Fig. 2b. The basins considered are reduced to 158 and are focused mainly in the center of CONUS. The raw average correlation, however, between $W_{Nov30}$ (now determined using a revised optimal time scale of 45 days, based on a supplemental calibration performed over the 158 basins) and subsequent streamflow has increased further to 0.50 in this subset of basins. It appears from Fig. 2b that retrieval quality does have an impact on our results; the SMAP L2 retrievals that are presumed to be more accurate are indeed more strongly connected to subsequent streamflow.

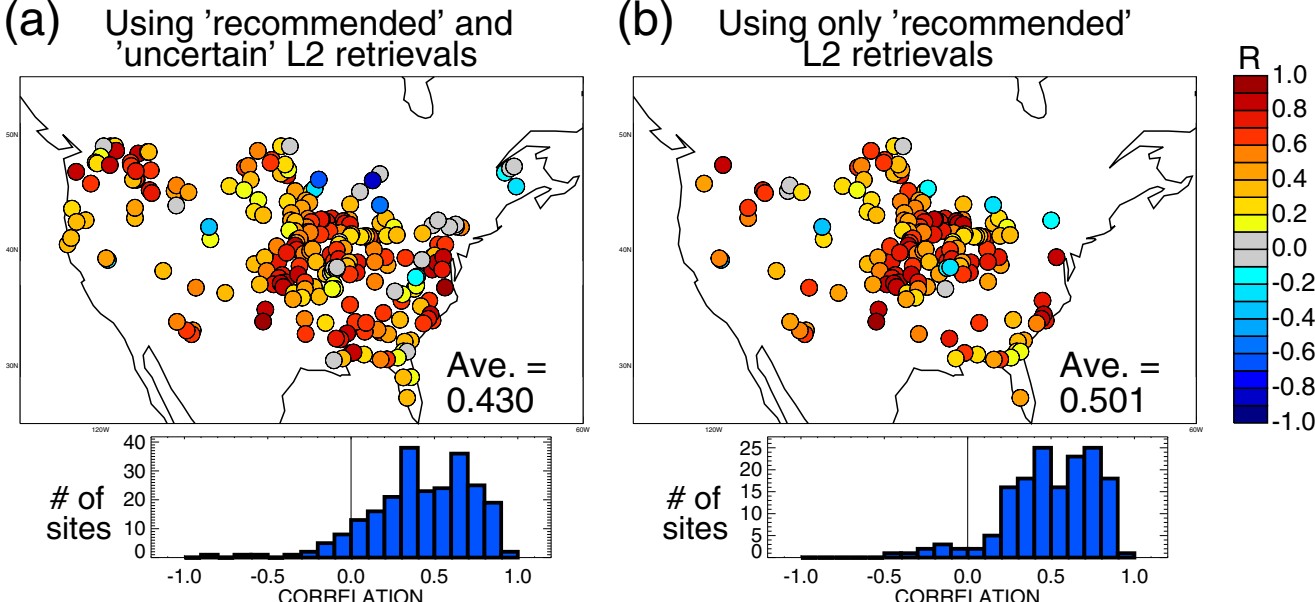

**Fig. 2 | Correlating November 30 soil moisture with subsequent February–May streamflow. a** Correlation between Soil Moisture Active-Passive (SMAP)-derived profile soil moistures ($W_{Nov30}$, as estimated from Level 2 [L2] surface moisture retrievals through Eq. (1)) and subsequent spring (February through May) streamflow amounts. For ease in visualization, values are plotted with circles of uniform size at the stream gauge locations. The attached histogram provides an overall summary of the results. **b** As in (**a**), but using only SMAP L2 retrievals flagged as being recommended. The values of $\tau$ underlying the calculations for the two panels are 38 and 45 days, respectively.

## Discussion

Using an exponential filter, SMAP L2 soil moisture retrievals during the falls of 2015–2021 are processed into November 30 estimates of profile soil moisture. As shown in Fig. 2, these late-fall estimates correlate significantly with accumulated streamflow measurements in the subsequent spring (February through May) across a wide variety of hydrological basins within CONUS. Positive correlations prevail throughout CONUS despite the complexity associated with streamflow generation processes—despite strongly region-specific impacts of topography, soils, snowpack, ecosystems, groundwater, freeze-thaw conditions, and other facets of the system on streamflow production. We emphasize that our goal here is not to parse out how elements of this complexity feed into or limit the correlations on a region-to-region basis; rather, it is simply to show that, even in the face of all this complexity, positive correlations appear across the entire study area—there is a real connection between autumnal SMAP data and subsequent springtime streamflow observations, a connection that spans the continent despite regional variations in hydrological processes. While some negative correlations do appear—this is inevitable given the small sample sizes examined and, for example, the vagaries of springtime precipitation variability (see, e.g., Supplementary Fig. S3)—the correlations in Fig. 2 could not be so overwhelmingly positive if a real connection did not exist.

The established connection, of course, has direct relevance to the prediction of springtime streamflow at the seasonal lead. A simple prediction model based on the linear regressions underlying the calculated correlations in Fig. 2 would provide levels of correlation-based prediction skill that are relatively high compared to those typically attained in the general realm of seasonal prediction, e.g., those obtained when operational seasonal forecast systems predict seasonally averaged midlatitude air temperature and precipitation at multimonth leads (https://www.cpc.ncep.noaa.gov/products/people/wwang/cfs_skills/). The raw average of the square of the correlations in Fig. 2a (after zeroing out negative values) is 0.27, meaning that on average, SMAP-predicted streamflow could explain over one-fourth of the variance of actual streamflow without any benefit from snow cover information or forecast precipitation. RMSE-based skill using such a linear regression model would similarly be non-negligible assuming that the statistical moments of basin streamflow are known a priori[10].

Of course, we could also evaluate in this regard the November 30 profile soil moisture estimates from MERRA-2—i.e., the reanalysis-based data we used to calibrate the exponential filtering of the SMAP data. Supplementary Fig. S4 shows that November 30 MERRA-2 soil moistures could provide a very similar amount of potential streamflow prediction skill; correlations are again essentially positive across CONUS, with an average value of 0.42. These similar results and the public's easy access to MERRA-2 data raise an obvious question: what advantages do SMAP data offer over reanalysis products for seasonal streamflow prediction? The key advantage offered by the SMAP data is accurate soil moisture information in otherwise poorly instrumented regions[11,12]. We should expect MERRA-2 soil moisture to be reasonably accurate over CONUS given the strong measurement networks there, particularly for precipitation, which drives the variability of soil moisture in the reanalysis. Across much of the globe, however, precipitation measurement networks are much less reliable, which will necessarily lead to less reliable MERRA-2 soil moisture estimates and thus a significantly reduced potential for streamflow prediction.

We can illustrate such reduced reliability through calculations using profile soil moistures generated with the NASA Global Earth Observing System Forward Processing for Instrument Teams (GEOS FPIT) product[13] (https://gmao.gsfc.nasa.gov/GMAO_products/NRT_products.php), which, like MERRA-2, is housed and operated by the NASA Global Modeling and Assimilation Office. In the context of this study, the GEOS FPIT system is essentially identical to the MERRA-2 system except for one key difference: in GEOS FPIT, rather than using observational rain gauge data to drive the land surface hydrology, the hydrology is driven instead with model-generated precipitation from the atmospheric analysis itself. The average correlation across CONUS between November 30 soil moistures from GEOS FPIT and subsequent springtime streamflow drops to 0.345 (see Supplementary Fig. S5), significantly less than that obtained with the SMAP-derived or MERRA-2 soil moistures. It is not a surprise that mostly positive values are still obtained with GEOS FPIT, given that rainfall in the FPIT system can still draw accuracy from other measurements, such as from aircraft and radiosondes, which have a higher-than-average density over CONUS. Nevertheless, not using direct rainfall measurements in GEOS FPIT has a clear detrimental effect, one that should reflect MERRA-2 performance in areas across the globe with limited or non-existent rain gauge information. In other words, the average of 0.42 found with the MERRA-2 soil moisture predictor over CONUS should not be expected to apply in many regions of the globe. In contrast, under the assumption that the time scale identified in Fig. 1 is itself widely applicable (a reasonable assumption given the high correlations seen in Fig. 1b across the many disparate climates of CONUS), the average of 0.43 found using the SMAP data (Fig. 2) should be representative even in regions without strong in-situ measurement systems, particularly in midlatitudes.

We emphasize midlatitudes in the previous sentence because we consider a prediction lead that spans the winter season to be particularly conducive to success—fall soil moisture anomalies may survive largely intact into the spring because ET during winter is greatly reduced and because snow falling after November 30 often may not melt and affect (or respond to) soil moisture conditions until the spring snowmelt season. In addition, in parts of midlatitudes[14–16], including central CONUS, precipitation rates are lower during winter, reducing potential variations in soil moisture. The potential for higher correlations during winter is supported by corresponding results in the study area for a lead that spans the summer season; the continental-scale average of the correlations in the summer case is, as expected, much lower (Supplementary Fig. S6). Latent predictability within the climate system, as indicated here for the winter lead, has been discussed extensively in the literature in the context of other climate processes[17,18]. (Note that we avoid December and January in any case because for many of the basins, SMAP L2 soil moisture retrievals are unavailable due to the presence of snow cover or frozen ground.) The implied predictability illustrated in Fig. 2 must mainly stem from soil moisture memory, given that the soil moisture anomalies examined here have a substantially larger connection (e.g., in terms of correlation) to the streamflow totals themselves than to the observed precipitation amounts leading into spring (the average correlation between $W_{Nov30}$ and December–May precipitation is only 0.23). Simply put, our results indicate that late fall soil moisture anomalies survive into spring and thereby determine the preconditioning of the springtime land surface for generating runoff.

The connection in Fig. 2a between the processed fall SMAP retrievals and subsequent spring streamflow (along with the associated implications for prediction) is the main result of the present study. A secondary result, however, is also worth highlighting. Two separate and independent calculations suggest that exponential filter-based averaging of the SMAP L2 soil moistures provides information on moisture deeper in the soil: (i) the correlations against the MERRA-2 profile soil moisture estimates in Fig. 1, and (ii) the fact that the averaged surface soil moisture does indeed provide information on subsequent streamflow (Fig. 2). The averages determined with a $\tau$ of 38 days are in fact more correlated with subsequent streamflow than are averages computed with shorter $\tau$ values (e.g., 10 or 20 days, see Supplementary Fig. S7). Our secondary result is thus consistent with past findings[19]; while soil moisture below a 5 cm depth is not directly accessible by the SMAP radiometer, information about this soil moisture nonetheless appears to be accessible through exponential

filtering. This result is also consistent with past studies demonstrating a connection between time-averaged surface moisture and deeper water storage variations[20].

Finally, we point out that the SMAP L4 product[21] combines SMAP brightness temperatures with observation-based meteorological data in a data assimilation framework, producing SMAP-based soil moisture estimates with ~2.5-day latency for both the root zone and the full soil profile that should, in principle (at least in areas with adequate meteorological forcing), be more accurate than those obtainable through the simple exponential filtering approach used here. Future work will include evaluating the L4 product for streamflow prediction and, in particular, parsing out the independent contribution of SMAP measurements (relative to that of the meteorological forcing used in the L4 product) to seasonal streamflow prediction accuracy.

## Methods

### Stream gauge data

We consider streamflow measurements for the spring period (February–May) of years 2016–2022 from 240 unregulated (unaffected by reservoir management) hydrological basins of intermediate size (2000–10,000 km$^2$) over CONUS, published by the United States Geological Survey (USGS) (https://waterdata.usgs.gov/nwis). The selected basins are a subset, based on basin area, of the 572 basins described by Kumar et al.[22]. The intermediate size was targeted because it is not smaller than the SMAP radiometer footprint but still small enough to allow most regions within CONUS to be represented. Of the 240 basins, 236 had streamflow data for all 7 years of the 2016–2022 period; we use these 236 basins for our analysis. The locations of the basins are indicated in Figs. 1 and 2; the circles in the panels are centered on the stream gauge locations.

### SMAP soil moisture retrievals

The SMAP Level 2 (L2) soil moisture retrievals are derived from space-borne L-band (1.4 GHz) radiometer measurements and represent average soil moisture conditions in the top several centimeters of soil[23]. The retrievals have been extensively evaluated against independent in-situ soil moisture data[24,25] and have been shown successful in meeting mission accuracy requirements. For the present study, we use the L2 Radiometer Half-Orbit Soil Moisture, version 8[26], baseline retrievals from the dual channel algorithm[27]. We use data collected on the descending branch of the SMAP orbit (6 AM local overpass time) and use only data flagged as having either "recommended" or "uncertain" quality, so that, for example, data collected during snow-covered periods are not considered. (We in fact consider the "recommended" subset of these data by themselves in Fig. 2b.) The data are analyzed on the 36-km Equal Area Scalable Earth grid[28] (version 2), which approximates the data's true underlying resolution.

At each of the 236 basins described in section "Stream gauge data" of Methods, we compute the basin-average near-surface soil moisture from the SMAP L2 retrievals. A basin average on a given day is considered valid only if the SMAP retrieval's grid cell area covers at least 50% of the basin area. These spatial averages are then time-averaged (see next section) to produce, for that basin, estimates of that year's late-fall profile soil moisture for correlation with subsequent springtime streamflow.

### Estimation of profile soil moisture

For the streamflow prediction problem, soil moisture estimates for a depth extending well below the top several centimeters of soil are particularly relevant, given that anomalies in this deeper moisture are more likely than surface anomalies to survive into spring. The first task in our analysis is thus to produce, using SMAP L2 surface soil moisture retrievals alone, estimates of such deeper-layer soil moisture. We will loosely refer to these L2-based deeper soil moisture estimates as profile moisture estimates, given that we calibrate the equation used to compute them with profile soil moisture estimates from an independent source.

Our chosen predictor for springtime streamflow is a basin's estimated profile soil moisture on November 30, $W_{Nov30}$. While this specific date is somewhat arbitrary—other dates in that general neighborhood produce similar results (see Supplementary Figs. S8 and S9)—it does satisfy our requirement of late-fall soil moisture for the analysis. A late-fall date is far enough from the spring snowmelt season to allow a demonstration of predictability at the seasonal time scale; furthermore, the fact that this seasonal lead spans the winter season allows us to take advantage of the added predictability (i.e., memory) associated with the quiescent nature of wintertime soil hydrology, a time when ET rates are low and much of the soil surface in North America is potentially frozen or covered by snow.

An established approach, the exponential filter[19,29,30], is applied to the surface soil moisture retrievals to obtain our estimates of deeper soil moisture. (Supplementary Fig. S10 illustrates the relevance of such temporal averaging.) In essence, if $t_{Nov30}$ is the day-of-year corresponding to November 30, we compute our estimate of basin-averaged profile soil moisture using:

$$W_{Nov30} = \sum_{n=0}^{N} w_n \exp\left(\frac{t_n - t_{Nov30}}{\tau}\right) / \sum_{n=0}^{N} \exp\left(\frac{t_n - t_{Nov30}}{\tau}\right) \quad (1)$$

where $t_n$ is the day-of-year lying $n$ days prior to November 30, $w_n$ is the basin-average of the soil moisture retrievals on day $t_n$, $N$ is a suitably large number of days, and $\tau$ is a chosen time scale. The discrete form of Eq. (1) allows the determination of $W_{Nov30}$ even under the intermittent availability of surface retrievals at the grid cell in question; this is of value given that the SMAP instrument has a typical return time of 2–3 days and may, in any case, feature data drop-outs.

For effective use of Eq. (1), we calibrate the time scale $\tau$ from available data—data that are, of course, independent of the streamflow data that we will eventually try to predict. Here we use reanalysis-based profile soil moisture estimates on November 30 (day 334 in non-leap years, day 335 in leap years) as the calibration target. In essence, through the calibration, we determine the single, universal value of $\tau$ that produces the greatest agreement between the basin-averaged retrieval data and these instantaneous profile soil moisture estimates from the reanalysis (which are also spatially averaged over each basin). The reanalysis used, the Modern-Era Retrospective analysis for Research and Applications, Version 2 (MERRA-2)[31], features a soil moisture hydrology driven by gauge-corrected precipitation forcing[32]; we use the profile soil saturation variable from the MERRA-2 holdings[33], which represents the average degree of soil saturation from the soil surface to the assumed bedrock. (This bedrock has a depth ranging from about 1.3 to 3.5 m in CONUS[34].) While the MERRA-2 estimates are far from perfect, they should have the first-order accuracy needed for this calibration exercise[35,36]. In any case, we show in the results section above that despite any MERRA-2 inaccuracies, the calibration successfully leads to soil moisture estimates of relevance for streamflow prediction.

In each of the 60 tests (one test for each value of $\tau$ evaluated, with imposed values ranging from 1 to 60 days), we use Eq. (1) to estimate November 30 profile soil moisture within each basin for every year during 2015–2021 from the SMAP retrievals and then compare these estimates to the corresponding MERRA-2 profile soil moisture. Agreement between the seven data pairs at each of the 236 basins is measured with the Pearson correlation coefficient, $R$. The overall calibration metric is then defined as the average of the 236 $R$ values. Note that we mitigate to some degree the deleterious effect of a small sample size by averaging these R values over hundreds of basins. The $\tau$ value found to produce the largest spatially averaged R is deemed optimal; this single calibrated value is used for all basins to translate

the basin-averaged SMAP L2 retrievals into $W_{Nov30}$ values for the streamflow prediction analysis.

### Soil moisture–streamflow correlations

At each basin, we calculate R between the seven values of $W_{Nov30}$ (one for each year in 2015–2021) and the measured basin streamflow accumulated over the subsequent February through May period (one value for each year in 2016–2022). Given that $W_{Nov30}$ does not utilize any information between November 30 and the streamflow period, this temporal correlation indicates the degree to which springtime streamflow totals can potentially be predicted months in advance from SMAP L2 soil moisture retrievals.

As in the time scale analysis, while a correlation derived from seven data pairs is naturally prone to significant sampling uncertainty, our analysis is made statistically tenable by the large number of basins considered here, basins that indeed span CONUS. We examine the distribution across the basins of the individual correlations along with their overall average, examining the degree to which they violate the null hypothesis of zero true correlation. At a given station, a correlation based on 7 data pairs is significantly different from zero with a confidence of 95% if it exceeds 0.68. This value was determined from a simple Monte Carlo exercise: 10,000 sets of 7 sample pairs, each element in each pair drawn from a random normal distribution, provided 10,000 correlation values, from which the null hypothesis value at the 95th percentile was determined.

In addition, to provide a confidence interval for the overall average as well as to curtail the ability of spatial correlations in streamflow or soil moisture to affect this average (i.e., to keep densely gauged areas from inappropriately dominating the statistics), we apply a clustering approach as used in previous studies[37,38]. The approach involves specifying a cluster radius (say, 2 degrees) and, through the use of a Fisher *Z* transformation, computing a 95% confidence interval (for the positive and negative sides separately) for the *R* value of each basin contained within a given cluster. The positive (negative) confidence intervals within that cluster are then averaged into a single confidence interval. Finally, the average of these cluster-specific values across the domain is computed and then divided by the square root of the number of clusters.

### Data availability

The Version 8 SMAP L2 retrievals are available from https://doi.org/10. 5067/LPJ8F0TAK6E0. USGS streamflow data are available from https:// waterdata.usgs.gov/nwis/dv/?referred_module=sw. MERRA-2 data are disseminated through the Goddard Earth Science Data and Information Services Center (GES DISC), with the particular data used here obtained at https://disc.gsfc.nasa.gov/datasets/M2T1NXLND_5.12.4/ summary. The November 30 soil moisture data (from SMAP, MERRA-2, and GEOS FPIT) and subsequent streamflow data used herein are available at https://doi.org/10.6084/m9.figshare.22593244.

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

## Acknowledgements

Funding for this work was provided by the NASA SMAP mission and the SMAP Science Team. Computational resources were provided by the NASA Modeling and Prediction (MAP) Program through their support of the NASA Global Modeling and Assimilation Office. We thank Sarith Mahanama for help with the datasets. USDA ARS is an equal-opportunity employer.

## Author contributions

R.D.K. oversaw the study and wrote the paper. Q.L. managed the datasets. All authors contributed significantly to the analysis and interpretation of the results as well as to the editing of the manuscript. These authors contributed equally: W.T.C., R.H.R.

## Competing interests

The authors declare no competing interests.
