## [Peer Review File · Nature Communications]

Late-fall satellite-based soil moisture observations show clear connections to subsequent spring streamflowREVIEWER COMMENTS

Reviewer #1 (Remarks to the Author):

General comments:

This is a simple but interesting manuscript demonstrating the correlation between late fall soil moisture as measured by the SMAP satellite and subsequent spring streamflow. Similar results have been shown, at smaller spatial scales, with in situ soil moisture data. The weakness of the present analysis is that only 7 years of SMAP data were available for the study. Thus the estimated correlation coefficients have high uncertainty. The study compensates for this weakness by including data from >200 basins, which increases the strength of the evidence. I think the manuscript could be improved by including time series of fall soil moisture and spring streamflow for a few example basins. I suggest such a plot for a representative basin with a strong positive correlation and for a basin representative of those with negative correlations. I think some discussion about the potential causes of the negative correlations should be included as well.

Specific comments:

line 112: Rephrase "flat regions". Consider "in both mountains and plains".

Reviewer #2 (Remarks to the Author):

Summary:

The authors compare exponentially time-averaged SMAP soil moisture leading up to November 30 to February-to-May accumulated streamflow for 236 intermediate-sized basins in the United States. The time decay of the exponential filter is determined by comparing the average correlation (across all basins) of the filtered SMAP soil moisture versus the soil profile-average soil moisture from the MERRA-2 reanalysis. The authors find that the November 30 filter SMAP soil moisture alone yields an average correlation of 0.43 with spring streamflow, not accounting for precipitation, snow, etc. The authors conclude that fall soil moisture contains some skill for seasonal forecasting of spring streamflow.

General comments:

The analysis in this manuscript is straightforward and has the potential to be understood and appreciated by a wide audience. The finding of skill for fall soil moisture derived from satellite observations to predict streamflow is noteworthy, but I can't decide if it was a significant result that was overlooked by many researchers, or simply one that is fairly obvious. I think most hydrologists would agree that fall soil moisture is important for predicting spring streamflow, but the authors of this manuscript have demonstrated that the predictive skill is also present in remotely sensed surface soil moisture (when filtered as the authors have done) and also put a number on that skill over a broad region with varying environmental settings. The authors also make a clear effort to fairly account for the limitations of the data (i.e. measures taken to address the fact that all R values are calculated from 7 points), and I see no issues with the analysis or methodology, and the analysis seems reproducible. The article is clearly written, but it could use some work to improve its flow. The jump from introduction to results was a bit jarring, for example. Some parts of the manuscript are also a bit repetitive, despite its short length. I think the figure quality also needs some work (use consistent text size, sharpen images, etc.).

Specific comments:

One piece that was missing from this manuscript was an analogous correlation analysis using the MERRA-2 profile average soil moisture on Nov 30. Given that MERRA-2 was used to calibrate the exponential filter, it is important to know whether MERRA-2 has better or worse predictive skill than filtered SMAP soil moisture. If SMAP is better, this would highlight the importance of satellite SM observations, and boost the novelty and impactfulness of this manuscript. If worse, then we can model/assimilate well enough that maybe SMAP is not the best tool for seasonal flow

prediction?

Figures: The authors have shown the results of the correlation analyses using a color scale applied to points located at the outlet of each watershed. However, the points overlap each other a lot, which obscures some of the results. Instead of points, could the authors instead plot the basin boundaries and fill with the same color scale, in order to solve the overlap problem? Given that the basin size (2000-10000 km²) seems to fall in between huc 7-8, maybe this would be possible?

REVIEWER COMMENTS

We thank the reviewers for their comments. Our responses are provided below (in blue italics).

Reviewer #1 (Remarks to the Author):

General comments:

This is a simple but interesting manuscript demonstrating the correlation between late fall soil moisture as measured by the SMAP satellite and subsequent spring streamflow. Similar results have been shown, at smaller spatial scales, with in situ soil moisture data. The weakness of the present analysis is that only 7 years of SMAP data were available for the study. Thus the estimated correlation coefficients have high uncertainty. The study compensates for this weakness by including data from >200 basins, which increases the strength of the evidence. I think the manuscript could be improved by including time series of fall soil moisture and spring streamflow for a few example basins. I suggest such a plot for a representative basin with a strong positive correlation and for a basin representative of those with negative correlations. I think some discussion about the potential causes of the negative correlations should be included as well.

We have added the requested time series, along with some discussion, in the Supplementary Material (with a pointer from the main text). Time series of soil moisture, streamflow, and precipitation are shown at three different sites, one site representing the average correlation of 0.43, one site with a higher correlation, and one site with a large negative correlation. The clear cause of the negative correlation (unpredictably high springtime precipitation) is indicated in the caption for that site; this issue is now spelled out in the main text along with a mention of another chief cause, namely, sampling issues associated with computing a correlation from 7 data pairs.

Specific comments:

line 112: Rephrase "flat regions". Consider "in both mountains and plains".

Revised to "...in both mountainous and non-mountainous regions..."

Reviewer #2 (Remarks to the Author):

Summary:

The authors compare exponentially time-averaged SMAP soil moisture leading up to November 30 to February-to-May accumulated streamflow for 236 intermediate-sized basins in the United States. The time decay of the exponential filter is determined by comparing the average correlation (across all basins) of the filtered SMAP soil moisture versus the soil profile-average soil moisture from the MERRA-2 reanalysis. The authors find that the November 30 filter SMAP soil moisture alone yields an average

correlation of 0.43 with spring streamflow, not accounting for precipitation, snow, etc. The authors conclude that fall soil moisture contains some skill for seasonal forecasting of spring streamflow.

This is indeed a nice summary of the paper.

General comments:

The analysis in this manuscript is straightforward and has the potential to be understood and appreciated by a wide audience. The finding of skill for fall soil moisture derived from satellite observations to predict streamflow is noteworthy, but I can't decide if it was a significant result that was overlooked by many researchers, or simply one that is fairly obvious. I think most hydrologists would agree that fall soil moisture is important for predicting spring streamflow, but the authors of this manuscript have demonstrated that the predictive skill is also present in remotely sensed surface soil moisture (when filtered as the authors have done) and also put a number on that skill over a broad region with varying environmental settings.

*Yes, this was indeed the point – to show that remotely sensed surface soil moisture *all by itself* contains information on subsequent streamflow in a wide variety of settings. This has important implications, for example, for streamflow prediction in regions without strong in-situ meteorological measurement systems. The text has been strengthened to emphasize this; see our response to the comment below regarding MERRA-2.*

The authors also make a clear effort to fairly account for the limitations of the data (i.e. measures taken to address the fact that all R values are calculated from 7 points), and I see no issues with the analysis or methodology, and the analysis seems reproducible.

Thank you. (No response necessary.)

The article is clearly written, but it could use some work to improve its flow. The jump from introduction to results was a bit jarring, for example. Some parts of the manuscript are also a bit repetitive, despite its short length.

We have added a segue between the introduction and results sections to improve the flow. We have also gone through the entire manuscript to limit superfluous repetition, though keeping in mind that some repetition is useful for driving home our main points.

I think the figure quality also needs some work (use consistent text size, sharpen images, etc.).

Sharper, final versions of the figures are being provided with the resubmission.

Specific comments:

One piece that was missing from this manuscript was an analogous correlation analysis using the MERRA-2 profile average soil moisture on Nov 30. Given that MERRA-2 was used to calibrate the exponential filter, it is important to know whether MERRA-2 has better or worse predictive skill than

filtered SMAP soil moisture. If SMAP is better, this would highlight the importance of satellite SM observations, and boost the novelty and impactfulness of this manuscript. If worse, then we can model/assimilate well enough that maybe SMAP is not the best tool for seasonal flow prediction?

A reasonable point. We address this comment with an additional figure (Supplementary Figure S4) and substantial added discussion in the main text:

“Of course, we could also evaluate in this regard the Nov. 30 profile soil moisture estimates from MERRA-2 – i.e., the reanalysis-based data we used to calibrate the exponential filtering of the SMAP data. Figure S4 of the Supplemental Material shows that Nov. 30 MERRA-2 soil moistures could provide a very similar amount of potential streamflow prediction skill; correlations are again essentially positive across CONUS, with an average value of 0.42. These similar results and the public’s easy access to MERRA-2 data raise an obvious question: what advantages do SMAP data offer over reanalysis products for seasonal streamflow prediction? One key advantage is that SMAP data offer valuable soil moisture information even in otherwise poorly instrumented regions (Dong et al. 2019; Reichle et al. 2021). We expect MERRA-2 soil moistures to be reasonably accurate over CONUS given the strong measurement networks there, particularly for precipitation, which drives the variability of soil moisture in the reanalysis. Across much of the globe, however, precipitation measurement networks are less reliable, which will necessarily lead to less reliable reanalysis soil moisture estimates and thus a significantly reduced potential for streamflow prediction. In other words, the average of 0.42 found with the MERRA-2 soil moisture predictor over CONUS presumably would not apply in most areas. In contrast, under the assumption that the time scale identified in Figure 1 is itself globally applicable (a reasonable assumption given the high correlations seen in Figure 1b across the many disparate climates of CONUS), the average of 0.43 found using the SMAP data (Figure 2) should be representative even in midlatitude regions without strong in-situ measurement systems. ”

Figures: The authors have shown the results of the correlation analyses using a color scale applied to points located at the outlet of each watershed. However, the points overlap each other a lot, which obscures some of the results. Instead of points, could the authors instead plot the basin boundaries and fill with the same color scale, in order to solve the overlap problem? Given that the basin size (2000-10000 km²) seems to fall in between huc 7-8, maybe this would be possible?

We have indeed spent much time trying to come up with the clearest way of showing our results – making our main findings clearly visible from a quick look at the plots. According to a simple calculation using our basin areas, the approach suggested by the reviewer would produce a map in which only about 1/8 of CONUS would show information, with the rest being whitespace; given the final sizes of the figures in the paper, this would make our main message less obvious. We tried reducing the size of the dots, but for the reduction of overlap to be significant enough to consider, the dots would need to be small, again leading to significant whitespace. After trying different approaches, we decided to retain the current graphical approach – it is certainly possible to see the values represented by dots that are partially covered by other dots, and the dots that do lie under other dots are chosen at random, so we are not improperly emphasizing good vs. bad results. And, of course, we show the CONUS-wide average in the plots as well as discuss in the text potential significance issues associated with the various clusters seen in the plots.

Having said all this, if the copy editors end up preferring a different means of illustrating our results graphically, we will naturally reconsider and work with them to modify the plots.

REVIEWER COMMENTS

Reviewer #1 (Remarks to the Author):

The revisions have addressed the reviewer comments and improved the manuscript. I have only a few further minor suggestions.

On line 80, define R.

On line 187, include also the influence of winter precipitation amounts, which are lower than precipitation amounts in other seasons in many locations. Where winter precip is significant, it is often frozen and not available for runoff or soil moisture recharge until spring. Thus it is not low ET alone that contributes to persistent winter soil moisture anomalies.

Line 196, smaller than what? Clarify the comparison.

Line 234, change "most of CONUS" to "most regions within CONUS".

Fig. S3. Investigate/comment on the possibility of negative correlations occurring because the soil was snow covered or frozen on or prior to Nov. 30. I notice the negative correlations were most prevalent in the northern part of the study area.

Reviewer #2 (Remarks to the Author):

I appreciate the authors' responses and the changes that they made to the manuscript.

However, I think one hurdle remains in this manuscript. The authors compared the predictive skill of their filtered SMAP fall soil moisture versus MERRA-2 fall soil moisture in correlation with spring streamflow, in order to see if SMAP outperforms MERRA-2. However, the results showed that the two are roughly equivalent. This somewhat makes sense, as the SMAP filter was calibrated to MERRA-2. The authors also make a valid argument that MERRA-2 is supplied with more extensive, and potentially more accurate, input data over CONUS compared to the rest of the globe. Thus, it is not unreasonable to expect that SMAP might outperform MERRA-2 in other parts of the world. However, the authors have not shown this in their manuscript. If the authors could show that SMAP outperforms MERRA-2 (e.g. for gaged basins in less instrumented parts of the globe, or even just one region), thus demonstrating that remotely sensed soil moisture provides an advantage over reanalysis products for seasonal streamflow prediction, then I think this manuscript would be novel and important enough to be accepted in a Nature-level journal. But until then, this point remains a presumption, not evidence.

One way that this might be accomplished is if the authors are able to find a method to decouple their filtered SMAP data from MERRA-2 altogether. It seems that there are many potential tau values for which the analysis performs well, so maybe it's not especially important to calibrate the SMAP data to MERRA-2 in the first place?

Lastly, I appreciate the authors' attempts to improve the overlapping of points in their maps. However, there must be a better way than the current visualization. Would jittering the points help? Or making the points a little smaller? Or maybe adding a separate histogram of the r values?

Please see our responses to the reviewers' comments below (in blue italics).

Reviewer #1 (Remarks to the Author):

The revisions have addressed the reviewer comments and improved the manuscript. I have only a few further minor suggestions.

On line 80, define R.

R is now explicitly defined in the text as the Pearson's correlation coefficient.

On line 187, include also the influence of winter precipitation amounts, which are lower than precipitation amounts in other seasons in many locations. Where winter precip is significant, it is often frozen and not available for runoff or soil moisture recharge until spring. Thus it is not low ET alone that contributes to persistent winter soil moisture anomalies.

We have added the underlined text:

"We emphasize midlatitudes in the previous sentence because we consider a prediction lead that spans the winter season to be particularly conducive to success – fall soil moisture anomalies may survive largely intact into the spring because evapotranspiration during winter is greatly reduced and because snow falling after Nov. 30 often may not melt and affect (or respond to) soil moisture conditions until the spring snowmelt season. In addition, in parts of midlatitudes^{15,16,17}, including central CONUS, precipitation rates are lower during winter, reducing potential variations in soil moisture."

Line 196, smaller than what? Clarify the comparison.

We have clarified the text as follows (new text underlined): "The implied predictability illustrated in Figure 2 must mainly stem from soil moisture memory, given that the soil moisture anomalies examined here have a substantially larger connection (e.g., in terms of correlation) to the streamflow totals themselves than to the observed precipitation amounts leading into spring (the average correlation between W_{Nov30} and December-May precipitation is only 0.23)."

Line 234, change "most of CONUS" to "most regions within CONUS".

Changed as suggested.

Fig. S3. Investigate/comment on the possibility of negative correlations occurring because the soil was snow covered or frozen on or prior to Nov. 30. I notice the negative correlations were most prevalent in the northern part of the study area.

We have added the following paragraph to the main text (at lines 116 forward): "Curiously, the few negative or very small correlations that do appear mostly lie in the northernmost parts of the study region. While some of the larger negative correlations reflect unexpected springtime precipitation extremes (Figure S3), we note that the more northern areas are often snow-covered in late November. Arguably, snowpack present on Nov. 30 represents water that could otherwise have infiltrated and modified the soil moisture before and up to that date; as a result,

we can speculate that late fall snowpack has the effect of degrading the connection between W_{Nov30} and subsequent streamflow. More analysis, of course, in conjunction with a longer data period would be needed to explain the correlations found at individual locations.”

Reviewer #2 (Remarks to the Author):

I appreciate the authors' responses and the changes that they made to the manuscript.

However, I think one hurdle remains in this manuscript. The authors compared the predictive skill of their filtered SMAP fall soil moisture versus MERRA-2 fall soil moisture in correlation with spring streamflow, in order to see if SMAP outperforms MERRA-2. However, the results showed that the two are roughly equivalent. This somewhat makes sense, as the SMAP filter was calibrated to MERRA-2. The authors also make a valid argument that MERRA-2 is supplied with more extensive, and potentially more accurate, input data over CONUS compared to the rest of the globe. Thus, it is not unreasonable to expect that SMAP might outperform MERRA-2 in other parts of the world. However, the authors have not shown this in their manuscript. If the authors could show that SMAP outperforms MERRA-2 (e.g. for gaged basins in less instrumented parts of the globe, or even just one region), thus demonstrating that remotely sensed soil moisture provides an advantage over reanalysis products for seasonal streamflow prediction, then I think this manuscript would be novel and important enough to be accepted in a Nature-level journal. But until then, this point remains a presumption, not evidence.

It won't surprise the reviewer that regions with useful streamflow data also tend to be regions with good precipitation measurements, making the identification of a large-scale “good streamflow data but poor input data” region difficult. However, we were able to come up with an appropriate and (we think) inspired way to directly address the reviewer's comment – we repeated the CONUS calculations using soil moisture from an analysis (otherwise equivalent to MERRA-2) that effectively uses degraded precipitation data (but still analysis-quality, reflecting expected MERRA-2 quality in poorly gauged regions) to force the land surface. The text now describes these additional calculations (new text underlined):

“...We should expect MERRA-2 soil moistures to be reasonably accurate over CONUS given the strong measurement networks there, particularly for precipitation, which drives the variability of soil moisture in the reanalysis. Across much of the globe, however, precipitation measurement networks are much less reliable, which will necessarily lead to less reliable MERRA-2 soil moisture estimates and thus a significantly reduced potential for streamflow prediction.

We can illustrate such reduced reliability through calculations using profile soil moistures generated with the NASA Global Earth Observing System Forward Processing for Instrument Teams (GEOS FPIT) product¹⁴ (https://gmao.gsfc.nasa.gov/GMAO_products/NRT_products.php), which, like MERRA-2, is housed and operated by the NASA Global Modeling and Assimilation Office. In the context of this study, the GEOS FPIT system is essentially identical to the MERRA-2 system except for one key difference: in GEOS FPIT, rather than using observational rain gauge data to drive the land surface hydrology, the hydrology is driven instead with

model-generated precipitation from the atmospheric analysis itself. The average correlation across CONUS between Nov. 30 soil moistures from GEOS FPIT and subsequent springtime streamflow drops to 0.345 (see Figure S5), significantly less than that obtained with the SMAP-derived or MERRA-2 soil moistures. It is not a surprise that mostly positive values are still obtained with GEOS FPIT, given that rainfall in the FPIT system can still draw accuracy from other measurements, such as from aircraft and radiosondes, which have higher-than-average density over CONUS. Nevertheless, not using direct rainfall measurements in GEOS FPIT has a clear detrimental effect, one that should reflect MERRA-2 performance in areas across the globe with limited or non-existent rain gauge information. In other words, the average of 0.42 found with the MERRA-2 soil moisture predictor over CONUS should not be expected to apply in many regions of the globe. In contrast, under the assumption that the time scale identified in Figure 1 is itself widely applicable (a reasonable assumption given the high correlations seen in Figure 1b across the many disparate climates of CONUS), the average of 0.43 found using the SMAP data (Figure 2) should be representative even in regions without strong in-situ measurement systems, particularly in midlatitudes.”

One way that this might be accomplished is if the authors are able to find a method to decouple their filtered SMAP data from MERRA-2 altogether. It seems that there are many potential tau values for which the analysis performs well, so maybe it's not especially important to calibrate the SMAP data to MERRA-2 in the first place?

Figure S7 (formerly S6) indeed shows that any tau value greater than, say, 30 days would give comparable results, so yes – we could avoid mention of MERRA-2 altogether. Still, choosing a value greater than 30 rather than a smaller value seems a little arbitrary. We feel that the approach described above, using FPIT data, already addresses reasonably well the issue at hand.

Lastly, I appreciate the authors attempts to improve the overlapping of points in their maps. However, there must be a better way than the current visualization. Would jittering the points help? Or making the points a little smaller? Or maybe adding a separate histogram of the r values?

We address this comment by adding the suggested histograms to the plots. The histograms do help convey the message more clearly, and we thank the reviewer for the suggestion. Note that we did try making the points smaller, but this had no obvious impact on the figure's message – the smaller points, while overlapping less, were more difficult to see. Jittering the points might be possible but is a bit unsatisfying given that the points, as currently plotted, indicate true stream gauge locations.

REVIEWERS' COMMENTS

Reviewer #2 (Remarks to the Author):

The authors have sufficiently addressed my comments from the previous revision. I only have a couple of minor comments, which the authors could easily address.

Using the GEOS FPIT analysis, I think the authors have done enough to suggest that the SMAP data have a potential (albeit not realized) advantage over MERRA-2 in areas with sparse or poor quality precipitation observations. Therefore, filtered SMAP data may actually provide useful skill in predicting spring streamflow beyond the model/assimilation products currently available. I suggest that the authors emphasize this in the abstract, as it is one of the main novel aspects of the manuscript. Something along the lines of:

"We thus show that the satellite-based soil moisture retrievals, all by themselves, have the potential to produce skillful seasonal streamflow predictions several months in advance. In poorly instrumented regions, satellite-based soil moisture retrievals could provide an advantage over reanalysis products for streamflow prediction."

The authors should add a link and/or statement of availability for GEOS-FPIT in the "Data Availability" section.

Please see our responses to the final set of reviewers' comments below (in blue italics).

Reviewer #2 (Remarks to the Author):

The authors have sufficiently addressed my comments from the previous revision. I only have a couple of minor comments, which the authors could easily address.

Using the GEOS FPIT analysis, I think the authors have done enough to suggest that the SMAP data have a potential (albeit not realized) advantage over MERRA-2 in areas with sparse or poor quality precipitation observations. Therefore, filtered SMAP data may actually provide useful skill in predicting spring streamflow beyond the model/assimilation products currently available. I suggest that the authors emphasize this in the abstract, as it is one of the main novel aspects of the manuscript. Something along the lines of:

"We thus show that the satellite-based soil moisture retrievals, all by themselves, have the potential to produce skillful seasonal streamflow predictions several months in advance. In poorly instrumented regions, satellite-based soil moisture retrievals could provide an advantage over reanalysis products for streamflow prediction."

We have added the following (slightly more concise) statement to the abstract, underlined here: "We thus show that the satellite-based soil moisture retrievals, all by themselves, have the potential to produce skillful seasonal streamflow predictions several months in advance. In poorly instrumented regions, they could perform better than reanalysis soil moisture products in this regard."

The authors should add a link and/or statement of availability for GEOS-FPIT in the "Data Availability" section.

We have added the requested link.